# Molecular Characterisation of a Novel and Highly Divergent Passerine Adenovirus 1

**DOI:** 10.3390/v12091036

**Published:** 2020-09-17

**Authors:** Ajani Athukorala, Jade K. Forwood, David N. Phalen, Subir Sarker

**Affiliations:** 1Department of Physiology, Anatomy and Microbiology, School of Life Sciences, La Trobe University, Melbourne, VIC 3086, Australia; a.athukorala@latrobe.edu.au; 2School of Biomedical Sciences, Charles Sturt University, Wagga Wagga, NSW 2678, Australia; jforwood@csu.edu.au; 3Sydney School of Veterinary Science, University of Sydney, Camden, NSW 2570, Australia; david.phalen@sydney.edu.au; 4Schubot Exotic Bird Health, Texas A&M College of Veterinary Medicine and Biomedical Sciences, College Station, TX 77843-4467, USA

**Keywords:** *Adenoviridae*, *Atadenovirus*, *Passerine adenovirus-1*, genome organisation, evolution

## Abstract

Wild birds harbour a large number of adenoviruses that remain uncharacterised with respect to their genomic organisation, diversity, and evolution within complex ecosystems. Here, we present the first complete genome sequence of an atadenovirus from a passerine bird that is tentatively named *Passerine adenovirus 1* (PaAdV-1). The PaAdV-1 genome is 39,664 bp in length, which was the longest atadenovirus to be sequenced, to the best of our knowledge, and contained 42 putative genes. Its genome organisation was characteristic of the members of genus *Atadenovirus*; however, the novel PaAdV-1 genome was highly divergent and showed the highest sequence similarity with psittacine adenovirus-3 (55.58%). Importantly, PaAdV-1 complete genome was deemed to contain 17 predicted novel genes that were not present in any other adenoviruses sequenced to date, with several of these predicted novel genes encoding proteins that harbour transmembrane helices. Subsequent analysis of the novel PaAdV-1 genome positioned phylogenetically to a distinct sub-clade with all others sequenced atadenoviruses and did not show any obvious close evolutionary relationship. This study concluded that the PaAdV-1 complete genome described here is not closely related to any other adenovirus isolated from avian or other natural host species and that it should be considered a separate species.

## 1. Introduction

Adenoviruses are medium-sized, non-enveloped, linear, double-stranded DNA (dsDNA) viruses within the family *Adenoviridae* [1]. The family *Adenoviridae* contains five accepted genera [1]. One of these genera, the *Atadenovirus*, was added in 2002, to include adenoviruses that were previously assigned to the genus *Mastadenovirus*, but varied significantly based on genomic size, structure, genes, and gene arrangement. Originally all the viruses in the genus were thought to have an A+T content bias [2,3], but with the discovery of new species in this genus, it has been shown that the A+T bias is not a consistent feature of this genus [4,5,6]. The size of sequenced atadenovirus genomes ranges between 27 and 34 kb, and all have the characteristic inverted terminal repeat (ITR) found in all adenoviruses [7]. Atadenoviruses have a set of core genes shared with the other adenovirus genera plus genus-specific genes. A feature of adenoviruses is their ability to acquire genes from their hosts, bacteria, fungi, and other viruses. The Atadenoviruses appear to be particularly adept at this and all atadenoviruses sequenced to date contain five or more genes acquired from other organisms or whose origin is not known [5,8,9,10]. These genes are diverse in their function and appear to be lost as often as they are acquired as the atadenoviruses evolve.

Atadenoviruses have been detected in a diverse range of hosts, including birds [4,11,12,13,14,15,16,17,18], reptiles (order Squamata; lizards, snakes, and worm lizards), ruminants [8,9,15,19], marsupials [20,21], and a common tortoise [22]. Using a partial DNA polymerase gene sequence, recent studies also report the presence of a large number of novel atadenoviruses circulating in wild passerine species of birds in Australia and Europe and passerine species kept in aviculture collections [4,17,18]. Because of the limited sequence information for these passerine adenoviruses, there is still considerable uncertainty about their phylogenetic relationship to each other and other atadenoviruses. Additionally, only two avian atadenoviruses, the psittacine adenovirus 3 (PsAdV-3) and duck atadenovirus (DAdV), have been fully sequenced to date [11,23,24]. In this paper, we present the complete genome sequence of an atadenovirus from a passerine bird.

## 2. Materials and Methods

### 2.1. Source of Sample and Extraction of DNA

An Eastern spinebill (*Acanthorhynchus tenuirostris*) was caught in a mist net as part of a fauna survey in the Windsor Downs Reserve Nature (33°39′0.42″ S, 150°48′31.97″ E), New South Wales (Australian Bird and Bat Banding Scheme (ABBBS) Banding Authority Number 1893, ABBBS project approval—cooperative project 8529, New South Wales National Parks and Wildlife Service—Scientific License No. SL101929). Droppings from the cloth bag that it was held in prior to banding were collected. The total genomic DNA was extracted from collected droppings, using a commercial kit (PurelinkTM Genomic DNA Mini Kit, Invitrogen, California, CA, USA) according to the manufacturer’s instructions. Initially, the extracted DNA was screened for the presence of adenoviruses, using an established PCR protocol that has been widely used to detect adenoviruses, and, based on a short amino acid sequence of the DNA polymerase gene, it was found to be an atadenovirus whose sequence clustered with atadenoviruses derived from other Australian passerine species [4].

### 2.2. Library Construction and Sequencing

The PCR positive gDNA was used for next-generation sequencing. The library preparation and sequencing was performed as described previously [25]. Briefly, the paired-end library with an insert size of 150 bp was prepared, using the QIAseq FX DNA Library Kit (Qiagen), starting with 10 ng of total genomic DNA (gDNA), as measured by Qubit (Invitrogen). Fragmentation, end repair, and A-addition of gDNA was performed in a mixture containing 5 µL FX buffer, 10 µL FX enzyme mix with 10 ng of gDNA, and the reaction mixture was run in a pre-chilled thermal cycler, with the following temperature cycling profile: 4 °C for 1 min, 32 °C for 8 min, 65 °C for 30 min, and 4 °C for an indefinite time. When the temperature reached 4 °C, 45 µL of ligation master mix was immediately added, and the reaction incubated at 20 °C for 15 min, followed by cleaning of the adapter ligation reaction, using JetSeq™ Clean (Bioline, London, United Kingdom) according to the protocol described in QIAseq FX DNA Library Kits. The purified library was PCR amplified and cleaned to remove PCR-generated adaptor-dimers, using JetSeq™ Clean (Bioline, London, United Kingdom) according to the protocol described in QIAseq FX DNA Library Kits, with final elution in 23 μL of buffer EB. The quality and quantity of the prepared library were assessed, using an Agilent TapeStation (Agilent Technologies, Mulgrave, Victoria, Australia) by the Genomic Platform, La Trobe University. The prepared library was normalised and pooled in equimolar quantities. Cluster generation and sequencing of the pooled DNA-library were sequenced on an Illumina^®^ NextSeq 500 platform (Illumina, San Diego, CA, USA), according to the manufacturer’s instructions, through the Australian Genome Research Facility, Melbourne.

### 2.3. Genome Assembly

Sequencing data were analysed according to a previously established pipeline [26,27,28], using CLC Genomics Workbench (version 9.5.4, CLC bio, a QIAGEN Company, Prismet, Aarhus C, Denmark) and Geneious (version 10.2.2, Biomatters, Ltd., Auckland, New Zealand). Briefly, a total of 11.7 million reads were generated from the paired-end library with 150 bp insert size, using Illumina^®^ NextSeq 500 (Illumina, San Diego, CA, USA) to obtain the complete genome of PaAdV-1. Preliminary quality evaluation for all raw reads was generated, pre-processed to remove ambiguous base calls and poor-quality reads, and trimmed to remove the Qiagen Universal adapter sequences. Trimmed sequence reads were filtered against the chicken genome (*Gallus gallus*, GRCg6a, GenBank: GCA_000002315.5) with an additional bird genome (*Apteryx australis mantelli*, GenBank accession number LK064674), to remove host DNA contamination. Moreover, reads were filtered against *Escherichia coli* bacterial genomic sequence (GenBank accession no. U00096), to remove amplified bacterial sequences. Unmapped reads were used as input data for de novo assembly, using SPAdes assembler (version 3.10.1) [29], under the “careful” parameter in LIMS-HPC cluster (La Trobe Institute for Molecular Science—High Performance Computing cluster, specialised for genomics research in La Trobe University). This yielded a single contig (>39.6 kbp) that corresponded to atadenovirus sequence, according to BLASTN searches of GenBank databases [30,31].

### 2.4. Genome Annotation and Bioinformatics

The assembled PaAdV-1 genome was first annotated, using the Genome Annotation Transfer Utility (GATU) [32] with Duck atadenovirus A (DAdV-A, GenBank accession no. AC_000004), as the reference genome and further verification of the predicted open reading frames (ORFs) were performed, using Geneious (version 10.2.2). In addition, several other atadenoviruses were used as reference genomes for the annotation process, to evaluate possible consequences of truncations or extensions at the N and C termini of predicted proteins and to compare ORFs with the genus of *Atadenovirus* orthologues rather than with a single reference genome. Open reading frames longer than 30 amino acids with minimal overlapping (overlaps cannot exceed 25% of one of the genes) to other ORFs were selected and annotated. These ORFs were subsequently extracted into a FASTA file, and similarity searches, including nucleotide (BLASTN) and protein (BLASTX and BLASTP), were performed on annotated ORFs as potential genes if they shared significant sequence similarity to known viral or cellular genes (BLAST E value  ≤ 10^−4^ or contained a putative conserved domain as predicted by BLASTX and BLASTP [33].

To predict the function of unique ORFs predicted in this study, the derived protein sequence of each ORF was searched by multiple applications, to identify conserved domains or motifs. Transmembrane (TM) helices were searched, using the TMHMM package v.2.0 (DTU Health Tech, Lyngby, Denmark) [34], HMMTOP [35], TMpred [36], and Geneious (version 10.2.2, Biomatters, Ltd., Auckland, New Zealand). Additionally, searches for conserved secondary structure (HHpred) [37] and protein homologs, using Phyre2 [38] and SWISS-MODEL [39], were used to help predict the function of unique ORFs predicted in this study. We also used SignalP v.5.0 [40] to identify possible signal peptides.

### 2.5. Comparative Genomics

The genetic organisation of the newly assembled PaAdV-1 genome with other selected atadenoviruses was visualized and compared, using CLC Genomic Workbench (version 9.5.4, CLC bio, a QIAGEN Company, Prismet, Aarhus C, Denmark). Comparative G+C (%) content and pairwise identity of representative atadenovirus species against PaAdV-1, on the basis of complete genome nucleotide sequences and similarity percentage of the selected core proteins sequences of atadenoviruses, were obtained by using Geneious software (version 10.2.2, Biomatters, Ltd., Auckland, New Zealand). Selected proteins were aligned, using MAFFT in Geneious (version 10.2.2, Biomatters, Ltd., Auckland, New Zealand), and percentage of protein sequences similarity was calculated under the following parameters: scoring matrix BLOSUM62, Gap open penalty = 1.53. Blosum62 with threshold 1 (percentage of residues which have score > = 1 in the Blosum62 matrix).

### 2.6. Phylogenetic Analyses

Phylogenetic analyses were performed, using newly assembled PaAdV-1 genome sequence determined in this study with other selected AdV genome sequences available in the GenBank database. Initially, 44 AdV genome sequences were selected from the representative AdV genera from where amino acid sequences of the four conserved genes, namely as DNA polymerase, pTP, penton, and hexon, were extracted individually. Both the individual and concatenated amino acid sequences of the selected four AdVs core genes were aligned separately with MAFTT (version 7.450), using G-INS-i (scoring matrix BLOSUM62; gap open penalty 1.53; offset value 0.123) algorithm implemented in Geneious (version 10.2.2, Biomatters, Ltd., Auckland, New Zealand) [41]. Individual sequences were annotated with the virus species name, followed by the GenBank accession number in parentheses. The initial neighbourhood joining (NJ) phylogenetic tree from the Global alignment was performed in Geneious tree builder with 1000 bootstrap replicates. Further maximum likelihood (ML)-based phylogenetic analysis was performed, using individual genes, and concatenated amino acids sequences alignment under LG substitution model with 500 non-parametric bootstrap replicates in PhyML implemented in Geneious (version 10.2.2, Biomatters, Ltd., Auckland, New Zealand) [42], but 100 non-parametric bootstrap replicates were chosen for individual genes. The final tree was visualised with FigTree (version 1.4.4) [43].

## 3. Results

### 3.1. Genome of PaAdV-1

The assembled passerine adenovirus 1 (PaAdV-1) complete genome was a linear double-stranded DNA molecule of 39,664 bp in length, which was the longest atadenovirus to be sequenced, to the best of our knowledge. Like most atadenoviruses, the PaAdV-1 genome contained a central conserved coding region bounded by two identical inverted terminal repeat (ITR) regions. The length of the ITR varies considerably in other atadenoviruses and ranges from 40 to 194 bp long [9,10]. The ITR of PaAdV-1 encompassed 193 bp each with the coordinates of 1–193 sense orientation and 39,450–39,642 antisense orientation. The novel PaAdV-1 genome sequence was shown to contain a balanced G+C percentage (53.70%). The known AdV genomes that were most closely related to the PaAdV-1, according to complete genome analysis, were psittacine adenovirus 3 (PsAdV-3; 55.58%), snake adenovirus 1 (SnAdV-1; 55.58%), bearded dragon adenovirus 1 (BDAdV-1; 55.20%), and duck adenovirus A (DAdV-A; 54.25%).

### 3.2. Genome Annotation and Comparative Analyses of PaAdV-1

The PaAdV-1 genome had 42 predicted methionine-initiated ORFs encoding proteins that were annotated as putative genes and were numbered from left to right (Figure 1 and Table 1). Comparative analysis of the protein sequences encoded by the predicted ORFs, using BLASTX and BLASTP, identified homologs with significant protein sequence similarity (E value ≤ 10^−4^) for 25 ORFs (Table 1), while 17 ORFs (ORF01-04, ORF6-18) were found to be unique according to the BLAST database. Among the predicted protein-coding ORFs of the PaAdV-1 genome, 24 were homologs to other AdVs gene products (Table 1). Among these homologues AdVs gene products, the highest number of protein-coding genes (21) in PaAdV-1 demonstrated homologs to the psittacine adenovirus-3 (PsAdV-3). The remaining three genes, encoding E1B-large T-antigen, E4, and a hypothetical protein (RH0, F-box related protein), were homologues to amniota adenovirus 1 (protein identity 25.26%, GenBank accession no. QEJ80749.1), DAdV-1 (protein identity 26.92%, GenBank accession no. AJA72340.1/), and BAdV-D (protein identity 30.15% GenBank accession no. NP_899151.1/), respectively. The gene product of ORF05 was predicted to encode a 92 amino acid (aa) (molecular weight/theoretical isoelectric point, Mw/pI-10.27 kDa/11.16). This protein was predicted to contain a protein homolog to prokaryotic ankyrin repeat domain-containing protein-50 by BLAST search with a >49% (query coverage 61% and E-value: 2.00 × 10^−5^) amino acid sequence similarity.

The orientation of the predicted conserved genes in the PaAdV-1 was identical to that of PsAdV-3 and DAdV-A (Figure 1). The left-hand (LH) region of the PaAdV-1 genome contained a genus *Atadenovirus* specific gene homologue of p32K (Figure 1), followed by genes encoding E1B small T-antigen and E1B large T-antigen that were homologues to PsAdV-3 and DAdV-A, respectively. The amino acid sequence similarity of p32K was relatively low, as compared to other atadenoviruses, ranging from 30.83% to 33.51%, where the highest similarity was demonstrated with SnAdV-1 (33.51). Open reading frames corresponding to conserved LH proteins present in other atadenoviruses were not found in PaAdV-1.

At the centre of the PaAdV-1 genome, all the expected AdVs conserved genes were found, and their degree of homology with other atadenoviruses is shown in Figure 1 and Table 1. The only expected gene that was not found was the U-exon gene. The maximum similarity of individual proteins of PaAdV-1 to homologs in other atadenoviruses varied significantly and was not predictable (Table 2). For example, the DNA polymerase and penton base protein showed the highest pairwise identity with homologous proteins from PsAdV-3, whereas hexon, DNA binding protein, and fibre protein displayed the highest match with DAdV-A. Among the major capsid proteins, fibre protein showed a low amino acid identity, ranging from 15.65% to 25.42%, whereas penton base and hexon exhibited a high amino acid identity with PsAdV-3 (75.94%) and DAdV-A (80.04%), respectively. Furthermore, the additional predicted conserved proteins analysed also demonstrated high identity with homologous atadenoviruses proteins from other species (Table 2).

In the right-hand (RH) region of PaAdV-1, ORFs corresponding to four E4 genes were found (Figure 1 and Table 1). Among them, three (E4.1, E4.2, and E4.3) exhibited the greatest amino acid homology with proteins from PsAdV-3 (protein identity ranging between 27% and 35%), whereas one (E4) had the greatest amino acid homology with proteins from DAdV-1 (26.92% amino acid identity). To the right of the E4 region, the PaAdV-1 genome contained only one ORF previously described in atadenoviruses. This ORF codes for a protein of 205 residues that appears to be a homolog of the F-box domain found in BAdV-D (protein identity 30.15%). The RH0 gene was followed by 13 ORFs whose predicted protein products have not been identified in other adenoviruses previously (Figure 1).

### 3.3. Unique ORFs in PaAdV-1

The PaAdV-1 genome encoded all the conserved genes present in other adenoviruses, except the U exon. Additionally, it contained 17 novel ORFs (ORF1-4 and ORF6-18, Table 1) that were not present in other adenoviruses sequenced to date, nor did they match sequences in the NR protein database, using BLASTP and BLASTX. These unique ORFs encoded proteins of 33–312 amino acids (aa) in length (Table 1). Among these, the novel ORF8 was predicted to encode a 39 aa-length protein (molecular weight/theoretical isoelectric point, Mw/pI-4.49 kDa/8.96). This protein was predicted to contain a protein homolog to hepatitis E virus capsid protein (PDB: c3ggqA) by Phyre2 with a 74% sequence coverage, but the confidence of predicted protein structure by phyre2 was quite low as 40%. Therefore, there was no good structure predicted by using Phyre2 and SWISS-MODEL.

Four novel ORFs (ORF09, -10, -13, and -18) were predicted to contain transmembrane helices (TMHs), but no classical signal peptide. ORF10 was predicted to encode a 312 aa protein (Mw/pI-35.60 kDa/6.35) containing at least two TMHs. The orientation of the protein in the TMHs is shown in Figure 2. Furthermore, the TMHs detected by EMBOSS in Geneious also showed the presence of alpha-helices (α-helices) within TMHs predicted region, which was dominated by highly hydrophobic residues (red colour) (Figure 2A). However, we were unable to model the structure of ORF10 and or TMHs by using the Phyre2, HHpred, and SWISS-MODEL; this might be due to the lack of closely related structure in the database. Though the function of TMHs and dominant hydrophobic residues in this novel ORF is unknown, studies have shown that hydrophobicity drives the insertion of helical segments into the transmembrane proteins and acts as a hallmark of soluble globular protein tertiary structure [44,45]. ORF18 was predicted to encode a 237 aa protein (Mw/pI-26.14 kDa/8.81) that was also predicted to have at least two TMHs (Figure 3). TMHs detected in ORF18 by EMBOSS in Geneious also showed the presence of α-helices within TMHs predicted region, and they were also shown to be dominated by highly hydrophobic residues (red colour) (Figure 3A). Similarly, ORF13 was shown to contain a single TMH by TMHMM, TMpred, and EMBOSS tool in Geneious used in this study (Appendix A). The protein encoded by a novel ORF09 (181 aa) was predicted to have at least one C-terminal TMH by several programs, including EMBOSS 6.5.7 tool charge in Geneious, TMHMM and TMpred (Appendix A). However, there was an additional N-terminal TMH detected by Geneious and TMpred, and a further TMH was predicted by TMpred (Appendix A). Nonetheless, there was no evidence for conserved secondary structure and or protein homologs detected by various software, including HHpred [37], (Phyre2) [38], and SWISS-MODEL [39].

### 3.4. Evolutionary Relationships of PaAdV-1

Phylogenetic reconstruction based on two non-structural (polymerase and pTP) and two structural (penton and hexon) protein sequences clearly supported the inclusion of the newly assembled PaAdV-1 in the genus *Atadenovirus*. In the resulting ML tree based on concatenated amino acid sequences of four selected AdVs genes, the novel PaAdV-1 occupied a distinct sub-clade position with strong bootstrap support, when compared with other sub-clades within the genus *Atadenovirus* (Figure 4), suggesting that it may represent an ancient evolutionary lineage within the genus. Using the same set of concatenated protein sequences, we found that the maximum inter-lineage sequence identity values between the novel PaAdV-1 and other atadenoviruses were >63% (PaAdV-1 vs. DAdV), >62.5% (PaAdV-1 vs. PsAdV-3), and >61.0% (PaAdV-1 vs. ruminant atadenoviruses), which mirrored the distinct phylogenetic position of this novel PaAdV-1. Furthermore, neighbour joining (NJ) phylogenetic inference of the concatenated protein sequences (Appendix A) and the ML trees based on individual protein sequences of the complete polymerase, penton, and hexon genes demonstrated similar tree topologies for the representatives of atadenoviruses species (Appendix A). For example, all of these ML trees based on individual genes showed that the novel PaAdV-1 was placed phylogenetically into a distinct sub-clade from all other atadenoviruses (supported by a strong bootstrap) and did not show any obvious close evolutionary relationship. However, ML phylogeny based on the amino acid sequences of the complete pTP gene supported the closest relationship of novel PaAdV-1 with reptilian and bird atadenoviruses (Appendix A).

## 4. Discussion

This paper describes the first complete sequence of an adenovirus from a passerine bird. Based on the organisation of its genes and closest homology to PsAdV-3 and DAdV-A, PaAdV-1 is most closely related to the atadenoviruses. However, PaAdV-1 is considerably more distantly positioned phylogenetically than the other sequenced atadenoviruses, forming a distinct sub-clade.

Interestingly, several unique ORFs in the PaAdV-1 genome were predicted to contain transmembrane helices, which are dominated by highly hydrophobic residues (Figure 2 and Figure 3 and Appendix A). Although the role of these ORFs in PaAdV-1 and its potential unique protein await characterisation, studies have shown that the hydrophobic residues of the transmembrane domains act as a hallmark to drive the creation of globular protein tertiary structure [44,45]. Furthermore, studies in human adenovirus have shown that transmembrane domains are required for the endoplasmic reticulum (ER) retrieval function to ensure efficient ER localisation and transport inhibition of MHC-I and MICA/B molecules [46].

PaAdV-1 was more similar to PsAdV-3 than any other atadenoviruses, by virtue of genome structure, complete genome sequence similarity (>55%), and G+C content (Table 2). The length of the ITRs of PaAdV-1 (193-bp) is significantly different from the ITRs of avian and ruminants atadenoviruses [9,11,47]; however, it seems to be similar in length to reptilian atadenoviruses [10]. Generally, the length of ITRs varies significantly throughout the family of *Adenoviridae,* being over 100 bp in the case of *Mastadenovirus, Ichtadenovirus*, and most of *Aviadenovirus*, while shorter than 60 bp in *Siadenovirus* and most of the non-reptilian atadenoviruses [9,10,48]. Although the function of these ITRs in most avian adenoviruses is largely unknown, studies have shown that ITRs from the various subgroups of human adenoviruses display robust transcriptional activity in a cell-type independent manner and actively reduce reporter gene expression [49]. To understand the biology of PaAdV-1 future experimentation would be desirable to isolate and propagate the virus. However, there is no established cell culture system to isolate PaAdV-1. Similar procedures adopted for the isolation of falcon adenovirus may represent a viable option for the isolation of PAdV-1, where the authors used several cell lines, including primary fibroblast cell lines from chicken, quail, and duck embryos [50].

The *Atadenovirus* genus was observed to contain overall lower genomic G+C percentage in the avian and ruminant viruses that were initially sequenced [2,9,23], with an exception for PsAdV-3, which was shown to contain a balanced G+C content (53.54%) [11]. Similar to the balanced G+C contents in reptilian atadenoviruses and PsAdV-3 genome sequence [5,10,11], the novel PaAdV-1 genome sequence also showed a balanced G+C content (53.70), in contrast to the G+C content seen with most of the of avian and ruminant atadenoviruses (Table 2). Interestingly, a very recent study has shown that the greatest diversity existed in the G+C contents in partial DNA polymerase gene of atadenoviruses that was originally sequenced from similar species of birds [4]. This scenario highlights that the low G+C content is not likely a consistent feature of the genus *Atadenovirus*, and it would be reasonable to take careful consideration to distinguish relatively closely related viruses based on G+C contents.

The most striking difference of the PaAdV-1 genome was documented at the right end region compared to any hitherto studied atadenoviruses, which included 14 unique protein-coding ORFs that were not present in any other adenoviruses sequenced to date. However, except for detection of transmembrane helices in several unique ORFs in the PaAdV-1 genome, the characteristics of these unique genes and their functionality with the host species was not fully understood. Like other studies, the presence of four E4 and one RH0 gene homologues in the right end region of PaAdV-1 genome clearly shows its affiliation with genus *Atadenovirus* [11,16,51]. In contrast to avian atadenoviruses such as PsAdV-3 [11] and DAdV-1 and -A [16,23], the PaAdV-1 contained a predicted RH0 gene which was shown to be homologue to an F-box domain-related hypothetical protein of BAdV-D [52]. The F-box domains are relatively rare among viral proteins, but they are extremely abundant in metazoans and are parts of the Skp1-cullin-F-box (SCF) E3-ubiquitin ligase complex, which is responsible for facilitating the degradation of a variety of critical regulatory proteins via SCF ubiquitin ligase complexes [53]. Although the potential role of the RH0 gene in PaAdV-1 is yet to be determined, it may appear to adopt a strategy to regulate a key cellular pathway that distinguishes some atadenoviruses from the other adenoviruses. This finding also highlights that the PaAdV-1 may follow a different cellular pathway than any other avian atadenoviruses, which requires further investigation.

After examining the phylogenetic inference between the novel PaAdV-1 and other AdVs, we see that it is evident that there was no obvious close evolutionary relationship; nonetheless, the PaAdV-1 was positioned distinctively in a sub-clade within the genus *Atadenovirus* clade (Figure 4 and Appendix A). In contrast to the previous studies where authors proposed the reptilian origin of atadenoviruses based on the comparison of phylogenetic trees of the adenoviruses, using a partial DNA polymerase gene [54,55], our resulting tress consistently demonstrated that atadenoviruses might have first evolved in a bird species that was present before the passerines and psittacine clades separated and was the ancestor of both clades (Figure 4 and Appendix A). The only exception, however, appears to be the phylogeny based on the pTP gene that supported the closest relationship of novel PaAdV-1 with reptilian and birds atadenoviruses (Appendix A). The highly divergent nature of the novel PaAdV-1 genome and its phylogenetic position likely indicated that the closest relatives of PaAdV-1 may yet to be uncovered, and the most common recent ancestor of this atadenovirus remains unresolved, which is consistent with many adenoviruses, including a recent atadenovirus [10] and aviadenovirus [56]. A similar scenario has been observed in a very recent study [4], where the authors highlighted that evolutionary relationship between viruses in the *Atadenovirus* genus appears to be unfolding and emerged as various lineages and sub-lineages that were not monophyletic, suggesting that they originated with a host-switching event that was followed by virus/species co-evolution. Therefore, to resolve the exact order of phylogenetic position and likely host switch events to birds, a further complete atadenoviral genome sequence from a diverse avian host species will be warranted.

## 5. Conclusions

This study reports the discovery of the first passerine adenovirus 1 (PaAdV-1) genome sequence from a passerine bird, Eastern spinebill. The novel PaAdV-1 genome sequence recovered in this study was highly divergent and provides insight into overall genome architecture that appears to be considered under a novel species, namely *Passerine adenovirus 1*, genus *Atadenovirus* and family *Adenoviridae*. The PaAdV-1 genome has enhanced the genomic information for the *Atadenovirus* genus and will shed new light in contributing to the better understanding of the atadenoviruses diversity and evolution. Moreover, it presents the possibility of examining the structure and function of its major structural and potential novel proteins. Further investigations into the sequencing of adenovirus genomes from other closely related species, as well as the pathogenesis, will broaden our understanding of host specificity for the adenovirus infection.

## Figures and Tables

**Figure 1 viruses-12-01036-f001:**
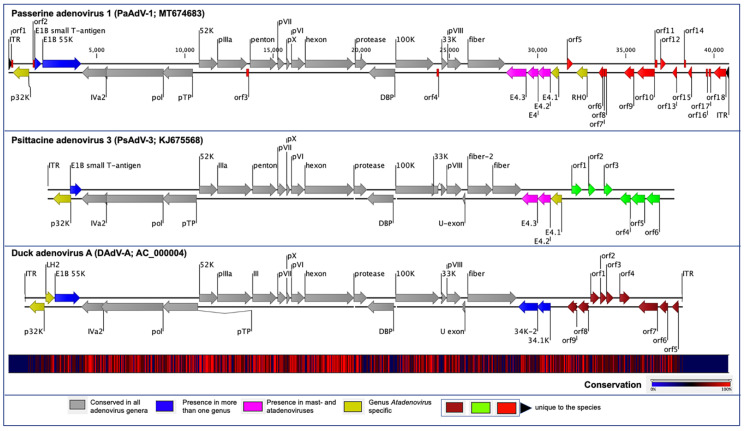
Schematic illustration of the avian atadenoviruses. Schematic map of the passerine adenovirus 1 (PaAdV-1, GenBank accession no. MT674683), in comparison with duck adenovirus A (DAdV-A, GenBank accession no. AC_000004) and psittacine adenovirus 3 (PsAdV-3, GenBank accession no. KJ675568), using CLC Genomic Workbench (version 9.5.4, CLC bio, a QIAGEN Company, Prismet, Aarhus C, Denmark). The arrows symbolize adenovirus genes and open reading frames (ORFs) predicted to code for proteins, indicating their direction of transcription. Each gene or ORF is colour coded, as indicated by the colour key in the legend. The bottom graph represents the sequence conservation between the aligned PaAdV-1, DAdV-A, and PsAdV-3 sequences at a given coordinate at each position in the alignment. The gradient of the colour reflects the conservation of that particular position is in the alignment. Red presents 100% conservation across all three viruses, black 50% conserved regions, and blue less than 50% conserved regions.

**Figure 2 viruses-12-01036-f002:**
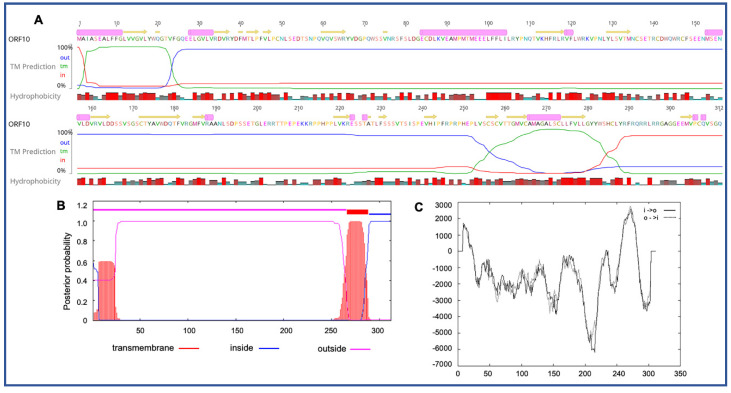
Predicted structure of the unique PaAdV1-ORF10. (**A**) prediction of transmembrane helices (TMHs) in unique PaAdV1-ORF10 gene using EMBOSS 6.5.7 tool in Geneious (version 10.2.2) (**A**), TMHMM (**B**), and TMpred (**C**). All the programs consistently predicted two TMHs. (**A**) TMHs detected by EMBOSS also showed the presence of alpha-helices within TMHs predicted region that has been dominated by highly hydrophobic residue (red colour). (**B**,**C**) The *x*-axis represents the position of residue, whereas *y*-axis represents the posterior probability (**B**), and scores (above 500 are considered significant) (**C**) for the predicted TMHs. (**C**) Solid and dashed black lines indicate protein orientation as inside to outside, and outside to inside, respectively.

**Figure 3 viruses-12-01036-f003:**
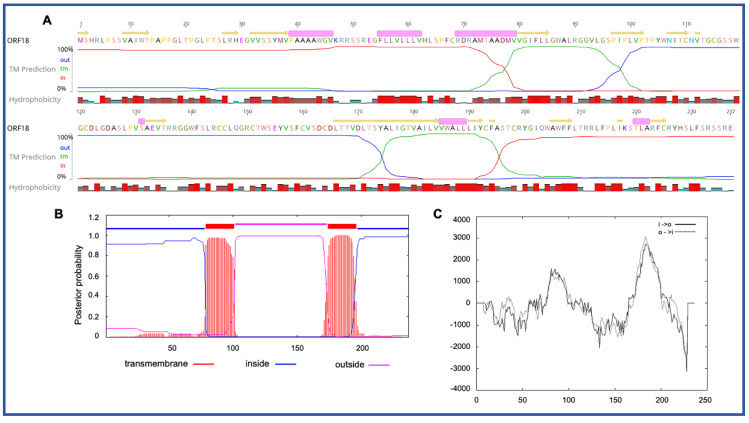
Predicted structure of the unique PaAdV1-ORF18. (**A**) prediction of transmembrane helices (TMHs) in unique PaAdV1-ORF18 gene, using EMBOSS 6.5.7 tool in Geneious (version 10.2.2) (**A**), TMHMM (**B**), and TMpred (**C**). All the programs consistently predicted two TMHs. (A) TMHs detected by EMBOSS also showed the presence of alpha-helices within TMHs predicted region that has been dominated by highly hydrophobic residue (red colour). (**B**,**C**) The *x*-axis represents the position of residue, whereas the *y*-axis represents the posterior probability (**B**) and scores (above 500 were considered significant) (**C**) for the predicted TMHs. (**C**) Solid and dashed black lines indicate protein orientation as inside to outside, and outside to inside, respectively.

**Figure 4 viruses-12-01036-f004:**
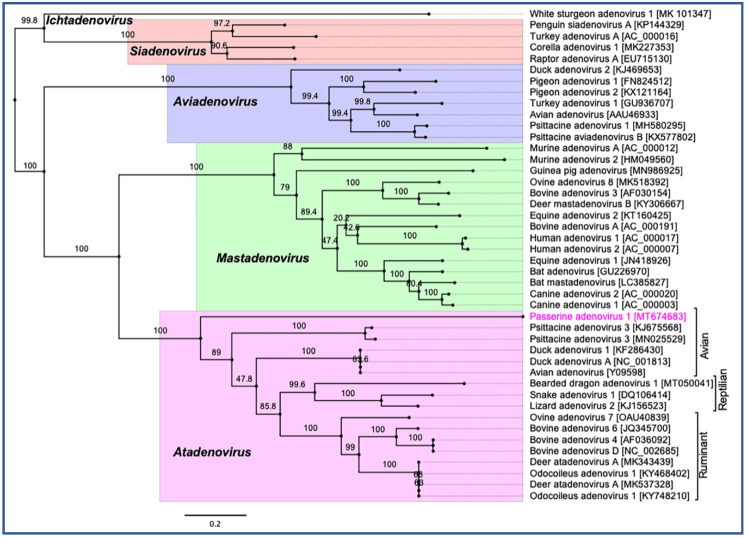
Phylogenetic tree shows the possible evolutionary relationship of novel passerine adenovirus 1 with other selected AdVs. Maximum likelihood (ML) tree was constructed by using concatenated amino acid sequences of the complete DNA-dependent DNA polymerase, pTP, penton, and hexon genes. Concatenated protein sequences were aligned with MAFTT (version 7.450) [41] in Geneious (version 10.2.2, Biomatters, Ltd., Auckland, New Zealand), under the BLOSUM62 scoring matrix and gap open penalty = 1.53. The gap >20 residues deleted from the alignments. The unrooted ML tree was constructed with PhyML [42] under the LG substitution model, and 1000 bootstrap re-samplings were chosen to generate ML trees, using tools available in Geneious (version 10.2.2, Biomatters, Ltd., Auckland, New Zealand). The numbers on the left show bootstrap values as percentages, and the labels at branch tips refer to original AdVs species name, followed by GenBank accession number in parentheses. The final tree is visualised with FigTree (version 1.4.4) [43]. The five official genera are highlighted as different background colours, and novel passerine adenovirus 1 is shown in pink colour.

**Table 1 viruses-12-01036-t001:** Predicted protein-coding genes of PaAdV-1.

PaAdV-1 Synteny	Start (nt)	Stop (nt)	Strand	Size (aa)	PsAdV-3 Synteny
ORF01	150	263	+	37	
p32K	250	1161	−	303	p32K
ORF02	1368	1469	+	33	
E1B protein, small T-antigen	1466	1849	+	127	E1B protein, small T-antigen
E1B protein, large T-antigen	1911	4049	+	712	
IVa2 protein	4094	5482	−	462	IVa2 protein
DNA polymerase	5164	8634	−	1156	DNA polymerase
pTP	8568	10,208	−	546	pTP
52K protein	10,547	11,581	+	344	52K protein
pIIIa protein	11,574	13,259	+	561	IIIa protein
ORF03	13,201	13,350	−	49	
penton protein	13,405	14,916	+	503	penton protein
pVII protein	14,942	15,289	+	115	pVII protein
pX protein	15,384	15,587	+	67	pX protein
pVI protein	15,651	16,403	+	250	pVI protein
hexon protein	16,421	19,150	+	909	hexon protein
Protease	19,221	19,901	+	226	protease
DBP	19,999	21,486	−	495	DNA-binding protein
100K protein	21,508	23,607	+	699	100K protein
ORF04	23,769	23,897	−	42	
33K protein	24,044	24,391	+	115	33K protein
pVIII protein	24,379	25,053	+	224	pVIII protein
fibre protein	25,407	27,509	+	700	fibre 2 protein
E4.3 protein	27,555	28,685	−	376	E4.3 protein
E4 protein	28,737	29,360	−	207	
E4.2 protein	29,299	30,036	−	245	E4.2 protein
E4.1 protein	30,058	30,513	−	151	E4.1 protein
ORF05	30,907	31,185	+	92	
RH0	31,341	31,958	−	205	
ORF06	32,572	32,808	−	78	
ORF07	32,762	32,902	−	46	
ORF08	32,896	33,015	−	39	
ORF09	33,901	34,446	−	181	
ORF10	34,604	35,542	−	312	
ORF11	35,541	35,672	+	43	
ORF12	35,828	36,118	+	96	
ORF13	36,429	36,683	−	84	
ORF14	37,082	37,195	+	37	
ORF15	37,306	37,527	−	73	
ORF16	38,325	38,432	−	35	
ORF17	38,483	38,611	−	42	
ORF18	38,767	39,480	−	237	

Notes: PaAdV-1, passerine adenovirus 1; PsAdV-3, psittacine adenovirus 3; aa, amino acid; nt, nucleotide.

**Table 2 viruses-12-01036-t002:** Comparative G+C (%) content and pairwise identity of representative atadenovirus species against passerine adenovirus 1 (PaAdV-1) on the basis of complete genome nucleotide sequences and selected core proteins amino acid sequences. Shading and bold front highlights maximum similarity. Selected proteins were aligned by using MAFFT in Geneious (version 10.2.2, Biomatters, Ltd., Auckland, New Zealand) with the chosen of following parameters to calculate % similarity: scoring matrix BLOSUM62, Gap open penalty = 1.53. Blosum62 with threshold 1 (percentage of residues which have score > =1 in the Blosum62 matrix).

	Reference Atadenoviruses	Genome Identity (%)	G+C Content	% Pairwise AA Similarity with PaAdV-1								
				p32K	IVa2	DNA Pol	pTP	Penton	Hexon	Protease	DBP	Fibre Protein
**Avian**	PaAdV-1		53.7									
	PsAdV-3	**55.58**	53.5	30.83	59.01	**53.45**	33.28	**75.94**	76.44	56.19	36.02	17.52
	DAdV-A	54.25	45.5	32.78	64.56	52.89	32.32	73.57	**80.04**	53.1	**36.97**	**25.42**
**Ruminant**	BAdV-D	53.38	37.1	29.73	**70.35**	52.73	32.37	72.69	76.09	56.19	35.77	22.38
	OAdV-7	53.69	33.6	33.53	69.97	51.71	32.77	73.29	77.89	53.98	36.07	23.27
	OdAdV-1	53.27	33.3	32.65	63.59	52.09	31.45	73.35	77.32	54.42	36.08	23.99
**Reptilian**	LAdV-2	53.43	47.8	32.47	59.23	52.39	**33.38**	72.91	76.64	56.64	35.45	15.65
	SnAdV-1	**55.58**	56.8	**33.51**	57.94	50.34	32.49	72.63	75.33	**58.41**	34.5	20.06
	BDAdV-1	55.2	56.6	33.15	58.80	50.70	33.14	69.18	75.84	58.08	31.72	24.35

PaAdV-1 (Passerine adenovirus 1, GenBank accession no. MT674683); PsAdV-3 (Psittacine adenovirus 3, GenBank accession no. KJ675568); DAdV-A (Duck adenovirus A, GenBank accession no. AC_000004); BAdV-D (Bovine adenovirus D, GenBank accession no. NC_002685); OAdV-7 (Ovine adenovirus 7, GenBank accession no. OAU40839); LAdV-2 (Lizard adenovirus 2, GenBank accession no. KJ156523); SnAdV-1 (Snake adenovirus 1, GenBank accession no. DQ106414); BDAdV-1 (Bearded dragon adenovirus 1, GenBank accession no. MT050041); AA, amino acid; highest identity/similarity values are highlighted with bold font and light grey background.

## Data Availability

The complete sequence of the PaAdV-1 genome was deposited in GenBank, under the accession number MT674683. Raw sequencing datasets that support the findings of this study are accessible through Mendeley Data repository via the following Link (http://dx.doi.org/10.17632/57zhhhs7xz.1) [57].

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
