# Peer review of "Molecular Characterisation of a Novel and Highly Divergent Passerine Adenovirus 1"

_viruses, 2020, doi:10.3390/v12091036_

Round 1
Reviewer 1 Report
In this manuscript, authors have sequenced genome of an adenovirus isolated from a passerine bird and presented the data regarding comparisons with genome sequence of other known adenovirus. Based on the genome analysis, authors claim that the identified adenovirus is the longest Atadenovirus sequenced till date.
Since Atadenoviruses have tendency to incorporate extra genes, the presence of 42 potential ORfs is not surprising.
The manuscript appears to be about determining another adenovirus genome sequence. Including the data regarding the structure and function of some of these unique ORFs will really increase the significance of the manuscript
Reviewer 2 Report
This paper describes for the first time the complete sequence of a passerine adenovirus genome.
The paper reads well and a thorough and competent analysis is presented in the paper. Although a large number of adenovirus genomes have been sequenced it is relevant to publish yet another one, particularly because this one seems to have some unique properties. A problem is the length of the paper in relation to its scientific significance.
The most striking finding is the excessive length of the genome and that no less than 17 novel open reading frames were identified. The question whether these new ORFs are expressed as proteins is left open. The fact that most of them are very short make their functional expression unlikely.
A worry for me is that an impure DNA source was used as the starting material. No attempts seem to have been made to enrich the DNA preparation for adenovirus DNA. The resulting sequences represent fragments from the bird genome and microbial genomes which were removed electronically. However, there must a surplus of sequences which would interfere with the assembly. The authors should consider this problem.
Specific questions:
Did the novel sequence match sequences in the psittacine and duck adenovirus genomes?
Which of the undescribed ORFs overlap with other ORFs
The figures, which deal with the structures of two hypothetical proteins should be deleted
Recommendation: I recommend publication of the paper after reduction by about 50% much space could be gained if the authors leave out Discussion and introduce some additional comments in the results section.
Reviewer 3 Report
The current article by Athukorala and colleagues present the sequence and characterization of a novel passerine adenovirus 1. The methods are in detail, the analysis are adequate, the discussion is also quite inspiring and comprehensive.
I just have a suggestion: although the virus isolation is necessary for further pathogen study, it is of course difficult. Can the authors add/discuss the potential cell culture system for such work? And moreover, without virus isolation, it is possible to clone the genome into workable plasmid, reviewed in (PMID: 28711987). Which may give a second option than virus isolation.
For me the fact is that, we can sequence and identify many virus in silicon, but without have them in culture is only dry exercise.
Another minor comment: the reference 49 is on AAV, seems not to do with adenovirus ITR.
